# Analysis of vehicle and pedestrian detection effects of improved YOLOv8 model in drone-assisted urban traffic monitoring system

**Huili Dou** *, **Sirui Chen, Fangyuan Xu, Yuanyuan Liu, Hongyang Zhao**

Zhejiang Institute of Communications, Hangzhou, Zhejiang, China

* douhuili3636@163.com

**Data availability statement:** The VisDrone 2019 dataset utilized in this study for object detection and tracking tasks is available on GitHub at https://github.com/VisDrone/VisDrone-Dataset. The code is available on GitHub at https://github.com/douhuili/VisDorone-YOLOv8.

**Funding:** This work was sponsored in part by Zhejiang Provincial Natural Science Foundation of China under Grant No. LTGG24E080005:

## Abstract

This study proposes an improved YOLOv8 model for vehicle and pedestrian detection in urban traffic monitoring systems. In order to improve the detection performance of the model, we introduced a multi-scale feature fusion module and an improved non-maximum suppression (NMS) algorithm based on the YOLOv8 model. The multi-scale feature fusion module enhances the model's detection ability for targets of different sizes by combining feature maps of different scales; the improved non-maximum suppression algorithm effectively reduces repeated detection and missed detection by optimizing the screening process of candidate boxes. Experimental results show that the improved YOLOv8 model exhibits excellent detection performance on the VisDrone2019 dataset, and outperforms other classic target detection models and the baseline YOLOv8 model in key indicators such as precision, recall, F1 score, and mean average precision (mAP). In addition, through visual analysis, our method demonstrates strong target detection capabilities in complex urban traffic environments, and can accurately identify and label targets of multiple categories. Finally, these results prove the effectiveness and superiority of the improved YOLOv8 model, providing reliable technical support for urban traffic monitoring systems.

## 1 Introduction

In modern urban traffic management, real-time and accurate vehicle and pedestrian detection is an important means to ensure traffic safety and efficiency. With the acceleration of urbanization, the increase in traffic flow has brought great challenges to traffic monitoring systems. Efficient traffic monitoring can not only reduce traffic accidents and improve road utilization efficiency, but also provide important data support for traffic planning and management. Therefore, developing an efficient and accurate vehicle and pedestrian detection algorithm has become a research hotspot [1,2].

Early vehicle and pedestrian detection methods mainly rely on traditional computer vision techniques, including background subtraction, edge detection, and statistical model-based methods. Background subtraction [3] is a classic target detection method that detects

Research on Data-Driven Service Performance Assessment and Maintenance Decision-Making of Rural Highway Bridge Groups.

**Competing interests:** The author(s) declared no potential conflicts of interest with respect to the research, authorship, and/or publication of this article.

foreground targets by subtracting background images from video sequences. Although this method is computationally simple, it performs poorly in complex scenes such as lighting changes, dynamic backgrounds, and shadows. Edge detection [4] uses edge information in the image to identify the target contour. Commonly used algorithms include Canny edge detection and Sobel operator. However, edge detection is sensitive to noise and has difficulty dealing with complex background and occlusion problems.

Methods based on statistical models, such as Gaussian Mixture Model (GMM) [5], detect foreground targets by establishing a probability model of pixel values. These methods perform well in static backgrounds, but their performance will drop significantly in scenes with dynamically changing backgrounds. In addition, the optical flow method is used to detect moving targets by analyzing the motion information between consecutive frames. This method is more effective in detecting moving targets, but there are certain challenges in terms of computational complexity and real-time performance.

With the development of computer hardware and image processing technology, machine learning methods have begun to be applied to the field of target detection. The method based on HOG (Histogram of Oriented Gradients) features and SVM classifier proposed by Dollár et al. has achieved remarkable results in pedestrian detection [6]. However, these methods rely on manually designed features and are difficult to cope with the changing actual scenes and complex backgrounds.

In recent years, with the rapid development of deep learning technology, object detection algorithms based on convolutional neural networks (CNN) have become mainstream, significantly improving the accuracy and efficiency of vehicle and pedestrian detection [7,8]. Deep learning methods are mainly divided into single-stage and two-stage object detection algorithms.

Two-stage detection algorithms first generate candidate regions and then perform fine classification and bounding box regression on these regions. R-CNN [9] is one of the earliest two-stage detection algorithms. Its basic idea is to first use selective search to generate candidate regions and then extract features and classify each candidate region. Although R-CNN performs well in detection accuracy, it has high computational complexity due to the need to perform separate convolution operations on each candidate region.

Fast R-CNN [10] significantly improved computational efficiency by sharing convolutional feature maps. Subsequently, Faster R-CNN introduced the Region Proposal Network (RPN) to further speed up the generation of candidate regions. Mask R-CNN [11] added a branch for instance segmentation based on Faster R-CNN, further expanding its application scope. Although two-stage detection algorithms perform well in accuracy, their computational complexity and inference time are long, making it difficult to meet real-time requirements.

In contrast, single-stage detection algorithms do well in speed by treating the object detection problem as a regression problem and directly performing object classification and bounding box regression on the entire image. The YOLO [12] series is a typical single-stage detection algorithm. YOLOv1[13] achieves end-to-end object detection by dividing the image into grids, each of which predicts a fixed number of bounding boxes and class probabilities. Although YOLOv1 performs well in speed, it has certain shortcomings in detecting small and dense objects.

YOLOv2 [14] introduced the anchor box mechanism to further improve the detection accuracy. YOLOv3 [15] introduced multi-scale feature fusion on this basis, making the model perform better in detecting objects of different scales. YOLOv4 and YOLOv5 have made improvements in network structure, training strategy and data enhancement, significantly improving the detection performance. As the latest version, YOLOv8 has significantly improved detection accuracy and speed, but in practical applications, YOLOv8 still faces

some challenges in complex urban traffic scenes, such as small object detection and multi-object occlusion.

To address the above issues, we propose an improved YOLOv8 model to improve its vehicle and pedestrian detection effect in urban traffic monitoring. Our improvements are mainly focused on the following aspects: First, by introducing a multi-scale feature fusion mechanism, the model's detection ability for targets of different scales is improved; second, an improved non-maximum suppression (NMS) algorithm is used to enhance the model's detection performance under multi-target occlusion; finally, data enhancement is used to further improve the model's generalization ability.

To verify the effectiveness of our method, we conducted experiments on public datasets: VisDrone 2019 [16]. The VisDrone 2019 dataset contains a large number of urban traffic videos shot by drones, with complex backgrounds and changing lighting conditions, and is an ideal dataset for testing the robustness of the model.

In summary, our main contributions are as follows:

1. Multi-scale feature fusion mechanism (MFF): By introducing a MFF mechanism, the performance of the model in detecting objects of different scales is improved, especially the detection capability of small objects.
2. Improved non-maximum suppression algorithm: The improved NMS algorithm is used to effectively solve the detection problem under multi-target occlusion and improve the detection accuracy.
3. Data enhancement: Through data enhancement, enabling it to maintain a high level of detection performance in different scenarios.

Through the research in this paper, we hope to provide an efficient and reliable vehicle and pedestrian detection solution for urban traffic monitoring systems, and further promote the development of intelligent traffic management.

## 2 Related work

In the field of vehicle and pedestrian detection, target detection algorithms can be roughly divided into single-stage, two-stage and other emerging methods.

### 2.1 Single-stage detection method

This method achieves end-to-end object detection and significantly improves the detection speed. However, YOLOv1 performs poorly in detecting small and dense targets and has limited detection accuracy for complex scenes. The design concept of YOLOv1 is to do only one forward propagation on an image and directly predict multiple bounding boxes and categories. This method simplifies the detection process and makes it advantageous in real-time applications. However, due to the fixed number of bounding boxes it predicts, it performs poorly when dealing with dense targets. YOLOv2 introduces the Anchor Boxes mechanism to further improve detection accuracy and improves the performance of the model by using Darknet-19 as the feature extraction network. The introduction of anchor boxes enables the model to better handle targets of different scales, thereby improving the robustness of detection. YOLOv2 also introduces Batch Normalization and high-resolution classifiers, which make the model more stable when trained on larger datasets. However, YOLOv2 still has challenges in dealing with small objects and occlusion. YOLOv3, based on YOLOv2, adopts multi-scale feature fusion, making the model perform better in detecting objects of different scales.

Specifically, YOLOv3 uses three layers of feature maps for prediction, each of which is dedicated to handling objects of different scales, thereby improving the overall detection performance. In addition, YOLOv3 also introduces logistic regression to handle classification tasks, reducing the possibility of misclassification. Although YOLOv3 performs well on multiple benchmark datasets, it still has room for improvement when dealing with extremely complex scenes (such as severe occlusion or low light conditions). YOLOv4 further improves the detection performance by integrating multiple technologies such as CSPDarknet53, SPP, and PAN. CSPDarknet53 reduces the amount of computation and improves the inference speed of the model by introducing cross-stage partial connections. SPP (Spatial Pyramid Pooling) enhances the receptive field of the model, and PAN (Path Aggregation Network) optimizes feature fusion. However, the complexity of YOLOv4 increases the difficulty of model training and deployment. YOLOv4 performs well in tasks dealing with multiple targets and complex backgrounds, but its high computational requirements limit its application in resource-constrained environments. The design of YOLOv5 emphasizes modularity and scalability, and improves detection performance by automatically learning the best anchor box and improving the loss function. However, there is still a certain detection error when dealing with multi-target occlusion and complex background. Although YOLOv5 performs well in practical applications, there is still room for improvement in detection accuracy in some extreme scenarios, such as rainy and foggy weather or nighttime monitoring. SSD [17,18] achieves efficient multi-scale target detection by performing detection on feature maps of different scales. Unlike YOLO, SSD makes predictions on multiple feature layers, which gives it an advantage in dealing with multi-scale targets. SSD uses default boxes to predict the location information of the target, which effectively improves the detection ability of the model. However, SSD performs poorly in small target detection and has limited processing capabilities for high-density target scenes. RetinaNet [19] proposed focal loss to solve the problem of imbalance between positive and negative samples and significantly improved the detection accuracy. Focal loss enhances the model's learning ability for difficult-to-classify samples by reducing the focus on easy-to-classify samples. RetinaNet uses ResNet as the feature extraction network and combines it with the FPN to perform detection at multiple scales. However, RetinaNet has a high computational complexity and is difficult to run efficiently in real-time systems. Although RetinaNet performs well in accuracy, its high computational requirements and inference time limit its widespread use in real-time applications.

The single-stage detection algorithm achieves a high detection speed by directly predicting the target location and category, which is suitable for real-time applications. However, these algorithms still have shortcomings in small target detection, complex background processing and multi-target occlusion. Our method aims to solve these problems and improve detection performance through multi-scale feature fusion, improved non-maximum suppression (NMS) algorithm and data enhancement technology.

## 2.2 Two-stage detection method

R-CNN generates candidate regions through selective search, and then extracts features and classifies each candidate region. Although R-CNN performs well in detection accuracy, it has high computational complexity and slow processing speed. The main problem of R-CNN is that it needs to perform convolution calculations on each candidate region separately, which leads to a large amount of repeated calculations and greatly limits its application scope. Fast R-CNN significantly improves computational efficiency by sharing convolutional feature maps. Specifically, Fast R-CNN first performs convolution operations on the entire image, and then performs ROI (Region of Interest) pooling on the generated feature map, thereby

reducing repeated calculations. However, the candidate region generation process is still time-consuming. Fast R-CNN improves detection speed by improving training and inference processes, but the efficiency of its candidate region generation is still a bottleneck. Faster R-CNN [19] introduces a region proposal network (RPN), which greatly speeds up the candidate region generation speed and improves the overall detection performance. RPN directly predicts the location and score of the candidate region by sliding a window on the convolutional feature map, thereby achieving end-to-end object detection. However, the model structure of Faster R-CNN is complex and difficult to train. Although Faster R-CNN performs well on multiple benchmark datasets, its complex network structure and high computational requirements limit its deployment in real-time applications. Mask R-CNN adds a branch for instance segmentation based on Faster R-CNN. This branch predicts pixel-level segmentation masks on each candidate region, thereby achieving multi-task learning of object detection and instance segmentation. Although Mask R-CNN performs well in object detection and segmentation tasks, it has large computational overhead and storage requirements. Although the multi-task learning of Mask R-CNN improves the accuracy and richness of detection, its high computational resource requirements limit its application in resource-constrained environments.

Two-stage detection algorithms improve detection accuracy by first generating candidate regions and then performing fine classification and regression on these regions. However, these algorithms usually have high computational complexity and slow processing speed. Our improved YOLOv8 model maintains a high detection speed while improving detection accuracy by introducing efficient feature fusion and optimized NMS algorithm, solving the computational bottleneck problem of two-stage detection algorithms.

## 2.3 Other methods

EfficientDet [20] achieves a good balance between detection performance and computational efficiency by optimizing the network structure and scaling strategy. EfficientDet uses EfficientNet as the backbone network and uses the compound scaling strategy to balance between different resolutions, depths, and widths to improve the detection capability of the model. However, the accuracy of EfficientDet decreases when dealing with extremely complex scenes. Although EfficientDet performs well in speed and accuracy, it still needs further optimization when dealing with extremely complex scenes (such as multi-target occlusion and complex background). YOLOv6 [21] is the latest development in the YOLO series. It further improves the detection performance by improving the network architecture and training strategy. YOLOv6 introduces a more efficient feature extraction network and an optimized loss function, and adopts a large number of data augmentation techniques to enhance the robustness of the model. However, there is still room for improvement when dealing with small targets and complex backgrounds. Although YOLOv6 performs well in practical applications, its detection accuracy still needs to be improved in some extreme scenarios. YOLOv7 [22] has made new breakthroughs in real-time object detection by introducing the Trainable Bag-of-Freebies mechanism. YOLOv7 further optimized the model's training strategy and data processing flow, making it perform well on multiple benchmark datasets. However, its robustness in multi-target occlusion scenarios still needs to be improved. YOLOv7 showed high detection accuracy when dealing with multiple targets and complex backgrounds, but its robustness and stability in extreme scenarios still need further study. Deformable DETR [23] achieved end-to-end efficient object detection by introducing deformable convolution and multi-head attention mechanism. Deformable DETR uses the Transformer [24] structure in object detection, captures global context information through the self-attention mechanism,

and introduces deformable convolution to adapt to objects of different shapes and scales. Although it performs well in detection accuracy, its training time is long and the computational complexity is high [25]. Although Deformable DETR performs well in detection accuracy and diversity, its high computational requirements and training time limit its widespread use in real-time applications. There are also some other methods [26–28] that have made outstanding contributions in autonomous driving detection. Unlike them, when dealing with occlusion and masking problems, we first enhance and preprocess the data to make it easier to be detected by the model, thereby improving data processing efficiency and achieving better results.

## 3 Method

To improve the effect of the YOLOv8 model on vehicle and pedestrian detection in urban traffic monitoring, we proposed an improved YOLOv8 model, which mainly includes three modules: MFF mechanism, improved non-maximum suppression (INMS) algorithm and data enhancement. Our network structure is shown in Fig 1.

### 3.1 Multi-scale Feature Fusion mechanism (MFF)

Fig 2 explains its structure in detail. The feature extraction network (such as CSPDarknet53) extracts feature maps of different scales from the input image. Assuming the size of the input image is H×W, the feature map output by the feature extraction network can be expressed as:

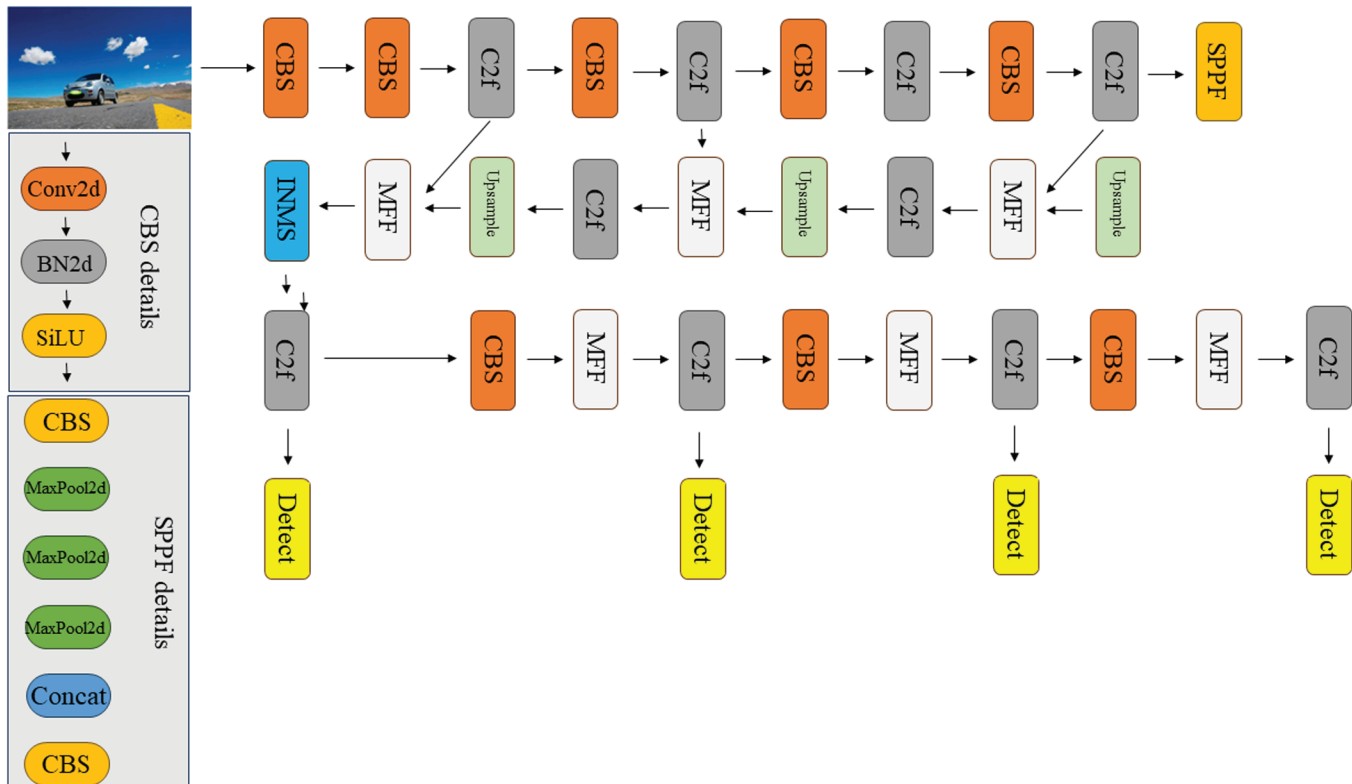

**Fig 1. Diagram of the proposed network structure.**

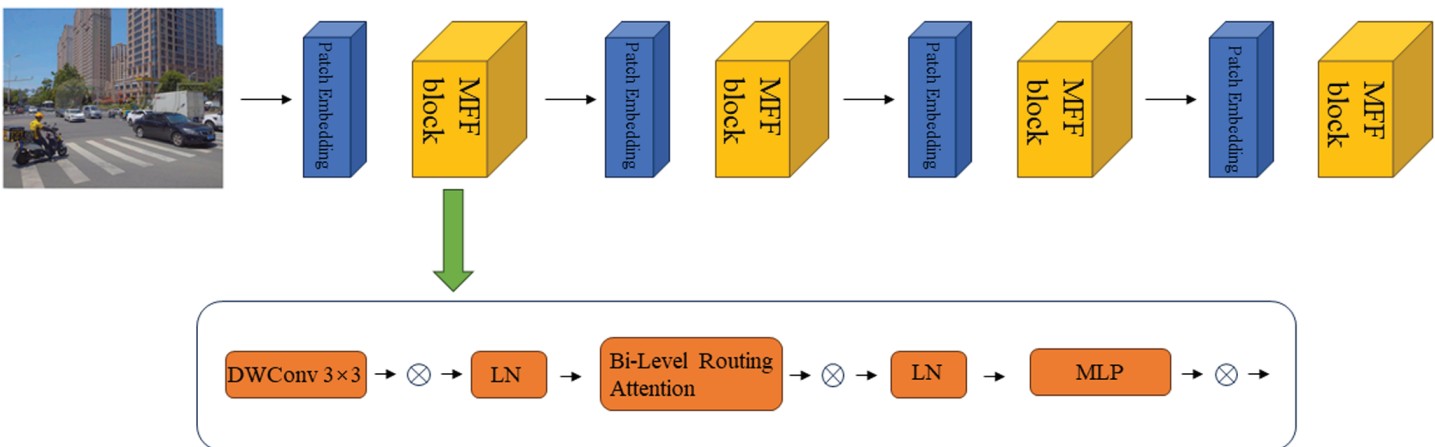

**Fig 2. MFF's overall structure.**

$F_i \in \mathbb{R}^{H_i \times W_i \times C_i}$, among them, $F_i$ represents the feature map of the $i$-th layer, $H_i$ and $W_i$ are the height and width of the feature map respectively, and $C_i$ is the number of channels.

The design and selection of the feature extraction network have an important impact on the final detection performance. The CSPDarknet53 network reduces the amount of computation and improves the efficiency of the network by introducing cross-stage partial connections. The network continuously extracts high-level features of the image by stacking convolutional layers and pooling layers layer by layer, and passes these features to the subsequent detection head for processing.

To achieve multi-scale feature fusion, we adopt the idea of Feature Pyramid Network (FPN) [24] to fuse feature maps of different scales. Specifically, assuming that the feature extraction network outputs three layers of feature maps $F_3$, $F_4$ and $F_5$, we first align the spatial size through upsampling operations and then fuse them through element-wise addition. The fusion process can be expressed as:

$$P_5 = F_5, \quad P_4 = F_4 + Upsample(P_5), \quad P_3 = F_3 + Upsample(P_4) \tag{1}$$

where $P_i$ represents the fused feature map, Upsample($\cdot$) represents the upsampling operation. By the way, we can obtain the fused multi-scale feature maps $P_3$, $P_4$ and $P_5$.

The main advantage of FPN is that it can effectively utilize the multi-scale information of the image. Traditional convolutional neural networks usually lose a lot of spatial information when processing high-level feature maps, while low-level feature maps contain rich spatial details but weak semantic information. By introducing FPN, we can improve the detection performance of the model.

The upsampling operation can be achieved through bilinear interpolation, as follows:

$$Upsample(F_i) = \sum_{m=1}^{H_i} \sum_{n=1}^{W_i} F_i(m,n) \cdot \delta\left(x - \frac{m}{H_i}, y - \frac{n}{W_i}\right) \tag{2}$$

among them, $\delta$ represents the Dirac delta function, $x$ and $y$ represent the position coordinates after upsampling.

The feature fusion process can be expressed as:

$$P_i = F_i + Upsample(P_{i+1}) \tag{3}$$

This layer-by-layer fusion method effectively combines feature information of different scales, so that the final feature map contains both high-level semantic information and low-level detail information, thereby improving detection accuracy.

After obtaining the fused multi-scale feature map, we need to generate the final feature map for detection. This step is usually achieved through convolution operations. Assuming that the weight of the convolution operation is $W_i$ and the bias is $b_i$, the generated feature map can be expressed as:

$$O_i = Conv(P_i, W_i) + b_i \tag{4}$$

where, $O_i$ represents the final generated feature map.

The process of feature map generation further extracts and refines feature information through a series of convolution operations. The convolution layer extracts information from local areas by sliding a window on the feature map and converts it into a high-dimensional feature representation. Through these operations, the generated feature map can better adapt to subsequent target detection tasks.

In order to better implement the multi-scale feature fusion mechanism, we considered the following points in model design:

1. Network structure optimization: Introduce cross-stage partial connection (CSP) and bottleneck structure in the feature extraction network to reduce the amount of calculation and improve the efficiency of the network.
2. Feature map upsampling: Bilinear interpolation is used to upsample the feature map to ensure that the upsampled feature map has sufficient spatial resolution and information transmission capability.
3. Fusion strategy: In this stage, different feature maps scales are effectively fused by adding them pixel by pixel to avoid information loss and redundancy.
4. Convolution operation: In the feature map generation stage, multi-layer convolution operations are used to further extract feature information to ensure that the final generated feature map has sufficient expressiveness and robustness.

Through the above improvements, our method can better combine detail information when processing objects of different scales, thereby improving detection performance. In particular, in terms of small target detection and complex background processing, the multi-scale feature fusion mechanism significantly improves the robustness and accuracy of the model.

## 3.2 INMS

To improve the accuracy of target detection, especially the performance in the case of multi-target occlusion, we propose an improved non-maximum suppression (NMS) algorithm. The NMS algorithm plays a key role in target detection. It reduces false detections by suppressing redundant overlapping boxes and retaining only the most likely candidate boxes. Although the traditional NMS algorithm is simple and effective, it is prone to false suppression when dealing with highly overlapping targets. The improved NMS algorithm we proposed solves this problem by introducing a weight mechanism and confidence adjustment.

**Traditional non-maximum suppression (NMS) algorithm.** Before introducing the improved NMS algorithm, let's review the traditional NMS algorithm. The steps of the traditional NMS algorithm are as follows:

1. Sorting: Sort all candidate boxes according to the confidence score output by the target detection network.
2. Select the highest confidence box: Select the candidate box with the highest confidence as the reference box.
3. Calculate the overlap rate: Calculate the overlap rate (IoU, Intersection over Union) between the reference box and other candidate boxes.
4. Suppression: Delete candidate boxes whose overlap with the reference box exceeds a certain threshold.
5. Repeat: Repeat the above steps until all candidate boxes are processed.

The above process can be expressed by the following formula:

$$IoU(A, B) = \frac{A \cap B}{A \cup B} \tag{5}$$

among them, $A$ and $B$ represent two candidate boxes respectively, $\cap$ represents the intersection of the two boxes, and $\cup$ represents the union of the two boxes.

Although the traditional NMS algorithm is simple and effective, it has obvious shortcomings when dealing with overlapping targets. When two targets are highly overlapped and have similar confidence levels, the traditional NMS algorithm may mistakenly suppress one of the targets, resulting in missed detection.

**Improved non-maximum suppression (INMS) algorithm.** In order to overcome the shortcomings of the traditional NMS algorithm, we proposed an improved NMS algorithm, which mainly includes the following steps:

1. Sorting and selecting the highest confidence box: Similar to traditional NMS, candidate boxes are first sorted according to their confidence, and the box with the highest confidence is selected as the reference box.
2. Calculate overlap rate and weight: Calculate the overlap rate (IoU) between the reference box and other candidate boxes, and calculate the weight of each candidate box based on the overlap rate.
3. Confidence adjustment: The confidence of the candidate boxes is adjusted according to the weight, so that the confidence of the candidate boxes with larger overlapping areas is reduced.
4. Suppress and update confidence: Delete candidate boxes and retain candidate boxes with higher confidence.

Specifically, the weight calculation formula of the improved NMS algorithm is:

$$w_i = \exp\left(-\frac{IoU(A, B_i)^2}{\sigma}\right) \tag{6}$$

among them, $w_i$ represents the weight of the $i$-th candidate box, $IoU(A, B_i)$ represents the overlap ratio between the reference box AAA and the iii-th candidate box $B_i$, and $\sigma$ is an adjustment parameter used to control the decay rate of the weight.

The confidence adjustment formula is:

$$C_i' = C_i \cdot w_i \tag{7}$$

among them, $C_i$ represents the original confidence of the $i$-th candidate box, and $C_i'$ represents the adjusted confidence.

By introducing a weight mechanism and confidence adjustment, the improved NMS algorithm can effectively suppress candidate boxes with large overlapping areas while retaining high-confidence candidate boxes, thereby reducing false positives and missed positives.

When implementing the improved NMS algorithm, we considered the following points: 1) Weight calculation: When calculating the weight of the candidate box, we use an exponential decay function to significantly reduce the weight of the candidate box with a large overlap rate, thereby reducing false detection. 2) Confidence adjustment: When adjusting the confidence, the product of the weight and the original confidence is used to further reduce the candidate box with low confidence, thereby enhancing the reliability of detection. 3) Threshold setting: During the confidence suppression process, an appropriate threshold is selected to ensure that the candidate box with low confidence is deleted while retaining the candidate box with high confidence, thereby improving the detection accuracy.

The INMS algorithm has the following advantages when dealing with multi-target occlusion and overlapping scenes: 1) Reduce false detection: Through the weight mechanism and confidence adjustment, the candidate boxes with large overlapping areas are effectively suppressed, reducing the occurrence of false detection. 2) Improve detection accuracy: In highly overlapping target scenes, the improved NMS algorithm can better distinguish different targets and improve detection accuracy. 3) Adapt to complex scenes: The improved NMS algorithm is more stable and more adaptable when dealing with complex backgrounds and multi-target occlusion.

By introducing the improved NMS algorithm, the target detection performance of our YOLOv8 model in urban traffic monitoring has been significantly improved, especially in multi-target occlusion and complex background scenes.

## 3.3 Data augmentation

Data augmentation is an important means to improve the generalization ability and performance of deep learning models. In response to the special needs of vehicle and pedestrian detection in urban traffic monitoring, we further introduced data augmentation technology based on the YOLOv8 model to improve the robustness and accuracy of the model in different scenarios. Data augmentation generates more training samples by performing a series of transformations on the training data to improve the generalization ability of the model. Specifically, we used the following data augmentation methods:

1. Rotation and flipping: Randomly rotate and flip images to increase the diversity of training data.
2. Scaling and Cropping: Randomly scale and crop the image to better adapt to objects of different sizes and positions.
3. Color transformation: Randomly adjust the contrast, brightness, saturation, and hue of the image to enhance the model's robustness to lighting changes.
4. Noise addition: Add Gaussian noise or salt and pepper noise to the image to improve the stability of the model in a noisy environment.

The application of these data augmentation methods can be expressed as follow, assuming that the original image is $I$ and the enhanced image is $I'$:

**Rotation and flipping.**

$$I' = Rotate(I, \theta), \quad I' = Flip(I) \tag{8}$$

among them, $\theta$ is the rotation angle, Rotate and Flip represent the rotation and flip operations respectively.

**Scaling and cropping.**

$$I' = Scale(I, s), \quad I' = Crop(I, (x, y, w, h)) \tag{9}$$

where $s$ is the scaling ratio, $(x, y, w, h)$ are the coordinates and size of the cropping window, Scale and Crop represent scaling and cropping operations, respectively.

Color Transformation.

$$I' = AdjustBrightness(I, \beta), \quad I' = AdjustContrast(I, \alpha) \tag{10}$$

$$I' = AdjustSaturation(I, \gamma), \quad I' = AdjustHue(I, \delta) \tag{11}$$

among them, $\beta$, $\alpha$, $\gamma$ and $\delta$ represent the adjustment parameters of brightness, contrast, saturation and hue respectively.

Noise addition.

$$I' = I + \mathcal{N}(0, \sigma^2) \tag{12}$$

where $\mathcal{N}(0, \sigma^2)$ represents Gaussian noise with mean 0 and variance $\sigma^2$.

## 4 Experiment

In this paper, we adopt a widely used datasets for object detection tasks: VisDrone 2019. The dataset is rich in diversity and complexity and are very suitable for verifying the effectiveness of our proposed improved YOLOv8 model for vehicle and pedestrian detection in urban traffic monitoring.

### 4.1 Dataset

**VisDrone 2019 dataset.** VisDrone 2019 [16] is a large-scale drone vision dataset released by the Institute of Automation, Chinese Academy of Sciences, mainly used for target detection, tracking and segmentation tasks. The dataset includes more than 33,000 images and more than 280 videos, covering complex urban scenes in different weather, time and environment. The VisDrone 2019 dataset consists of a training set, a validation set and a test set. The training set contains 10,209 images and 540,000 annotated boxes, the validation set contains 4,190 images and 220,000 annotated boxes, and the test set contains 6,889 images.

The dataset annotates 10 categories of targets, mainly including pedestrians, cyclists, cars, trucks, buses, motorcycles, bicycles, vehicles, traffic lights and traffic signs. The VisDrone 2019 dataset is highly diverse, covering different weather conditions (such as sunny, cloudy, rainy) and time periods (such as day, dusk, night), and the images contain a lot of complex backgrounds, such as buildings, trees and other pedestrians or vehicles, which increases the difficulty of target detection. In addition, many images contain densely distributed objects with many overlaps and occlusions, which places higher demands on the object detection algorithm.

In the experiment, we used the VisDrone 2019 dataset for model training and verification, and evaluated the performance of the model on this dataset to verify the detection capability of our proposed improved YOLOv8 model in complex urban scenes.

**Dataset preprocessing and preparation.** Before using VisDrone 2019 for experiments, data preprocessing and preparation are required. First, data cleaning is performed to remove duplicate and incorrect annotations in the dataset to ensure the quality of the training data. Then, the images are normalized to meet the standard size and format of the model input. Next, the aforementioned data augmentation methods (such as rotation, flipping, scaling, cropping, etc.) are applied to generate more training samples. Finally, the annotation files are converted into a format that the model can recognize to ensure that the coordinates and category information of the target box are correct.

VisDrone 2019 was selected for experiments mainly based on its diversity and complexity. The dataset covers a variety of traffic scenarios and complex backgrounds, which can fully verify the adaptability of the model in different environments. In addition, these two datasets are widely used in academia and industry. By conducting experiments on the dataset, our research can have higher reference value and application prospects. Finally, there are a lot of occlusions and overlaps in these two datasets, which put higher requirements on the target detection algorithm. Through experiments on these challenging datasets, the robustness of the model in dealing with complex scenes can be verified.

Through experiments at VisDrone 2019, we can comprehensively evaluate the performance of the improved YOLOv8 model in urban traffic monitoring and provide strong data support and technical guarantee for intelligent traffic management. The richness, diversity and complexity of these datasets make our experimental results more convincing and practical.

## 4.2 Evaluation metric

For the performance of the improved YOLOv8 model in urban traffic monitoring, we used a series of evaluation indicators, including model size, detection speed per image, precision, recall, F1-score, mAP0.5, and mAP0.5:0.95. These indicators can comprehensively measure efficiency of the model.

**Recall and precision.** Precision refers to the ratio of the number of positive samples correctly predicted by the model to all positive samples predicted. As follow:

$$Precision = \frac{TP}{TP + FP} \qquad (13)$$

among them, TP (True Positive) represents true positive examples, and FP (False Positive) represents false positive examples.

Recall refers to the ratio of the number of positive sample correctly. Its calculation formula is:

$$Recall = \frac{TP}{TP + FN} \qquad (14)$$

among them, FN (False Negative) represents false negative examples.

The relationship between precision and recall can be represented by a P-R curve, which shows the changes in precision and recall of the model under different thresholds.

**F1-score.** F1-Score is the harmonic mean of precision and recall, which is used to comprehensively evaluate the detection performance of the model. Its calculation formula is:

$$F1 = 2 \times \frac{Precision \times Recall}{Precision + Recall} \qquad (15)$$

F1-Score is widely used in detection tasks, especially on unbalanced datasets, where it can provide a more comprehensive performance evaluation.

**Average Precision (AP).** AP is the area under the P-R curve and can be calculated by the following formula:

$$AP = \int_0^1 P(R)dR \qquad (16)$$

where P(R) represents the precision at different recall rates.

**Mean Average Precision (mAP).** mAP is the average of APs of multiple categories and is used to measure the overall detection performance of the model on all categories. Its calculation formula is:

$$mAP = \frac{1}{N}\sum_{i=1}^{N} AP_i \qquad (17)$$

where $N$ represents the number of categories and $AP_i$ represents the AP value of the $i$-th category.

For mAP0.5, the IoU threshold is 0.5, that is, as long as the IoU between the predicted box and the true box is greater than 0.5, the detection is considered correct. For mAP0.5:0.95, the IoU threshold is gradually increased from 0.5 to 0.95 at intervals of 0.05, and the average AP under these thresholds is taken, which can more strictly evaluate the performance of the model.

Through the above evaluation indicators, we can comprehensively measure the vehicle and pedestrian detection performance of the improved YOLOv8 model in urban traffic monitoring. These indicators can not only evaluate the precision and recall rate of the model, but also reflect the overall performance of the model under different IoU thresholds, thereby providing a strong reference for practical applications.

## 4.3 Results and analysis

For the effectiveness of the improved YOLOv8 model we proposed, we conduct comparative experiments on the VisDrone2019 validation set (VisDrone2019-val). In this paper, we compared the detection accuracy of the improved YOLOv8 model with the original YOLOv8 model on multiple target categories. Table 1 shows the detection accuracy of the two models on different target categories (with mAP0.5 as the evaluation indicator).

The data in Table 1 shows that the detection accuracy of the improved YOLOv8 model on multiple target categories is better than that of the original YOLOv8 model. In terms of pedestrian detection, the accuracy of the improved YOLOv8 model is improved from 40.62% to 45.92%; in terms of vehicle detection, the accuracy of the improved model is improved from 78.82% to 81.52%. In addition, in the more complex tricycle and motorcycle detection, the improved YOLOv8 model also shows higher detection accuracy, from 26.82% and 43.52% to 32.82% and 48.22% respectively.

Overall, the improvement of the improved YOLOv8 model in mAP0.5 is more significant, from 38.52% to 43.32%. These results show that by introducing multi-scale feature fusion,

**Table 1. Comparison results between our method and yolov8.**

| Models | Bicycle | Tricycle | Pedestrian | Bus | Truck | Van | People | Awning-Tricycle | Car | Motor | mAP0.5 (%) |
|---|---|---|---|---|---|---|---|---|---|---|---|
| YOLOv8 | 40.62 | 32.22 | 12.42 | 78.82 | 43.62 | 34.42 | 26.82 | 15.22 | 57.32 | 43.52 | 38.52 |
| Ours | **45.92** | **35.72** | **16.42** | **81.52** | **49.12** | **42.42** | **32.82** | **17.72** | **62.92** | **48.22** | **43.32** |

improved non-maximum suppression (NMS) algorithm, and data enhancement and transfer learning strategies, our method shows higher detection accuracy and robustness in vehicle and pedestrian detection tasks in complex urban scenes.

Through the above experimental results, we have verified the superior performance of the improved YOLOv8 model in processing different target categories and complex backgrounds, providing a more efficient and reliable solution for urban traffic monitoring systems. This improvement not only proves the effectiveness of our improvement strategy, but also provides important reference value for future target detection research.

Fig 3 show the precision-confidence curves and precision-recall curves of the original YOLOv8 model and the improved YOLOv8 model on the VisDrone 2019 validation set. As can be seen from the figure, the precision and recall of the improved YOLOv8 model on most target categories are higher than those of the original YOLOv8 model. In the precision-confidence curve, the curve of the improved YOLOv8 model is generally higher than that of the original YOLOv8 model, especially in the categories of pedestrians, cars, tricycles and motorcycles. This shows that the improved model can maintain a higher precision at a higher confidence when detecting these targets.

In the precision-recall curve, the curve of the improved YOLOv8 model is also higher than that of the original model, indicating that the improved model can maintain a high recall while ensuring high precision. This further verifies that the improved model has better detection capabilities when dealing with complex backgrounds and multi-target occlusions.

Fig 4 show the F1-confidence curves and recall-confidence curves of the original YOLOv8 model and the improved YOLOv8 model. As can be seen from the figure, the F1 value and recall rate of the improved YOLOv8 model on each target category have been significantly improved. In the F1-confidence curve, the curve of the improved YOLOv8 model is higher than that of the original model in each confidence range, especially in the high confidence area, which shows that the improved model can maintain a higher F1 value at high confidence, that is, it combines the performance of precision and recall.

In the recall-confidence curve, the recall rate of the improved YOLOv8 model at different confidence levels is generally higher than that of the original model, especially in the low confidence area, the recall rate of the improved model is more significantly improved, which

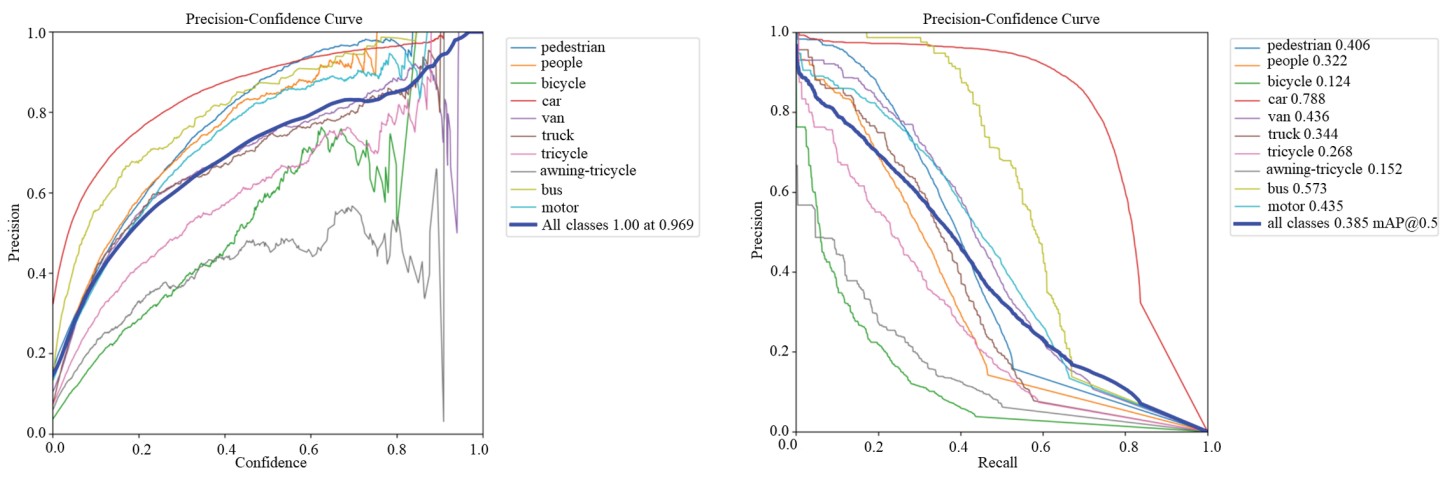

**Fig 3. Evalution factor of yolov8s.**

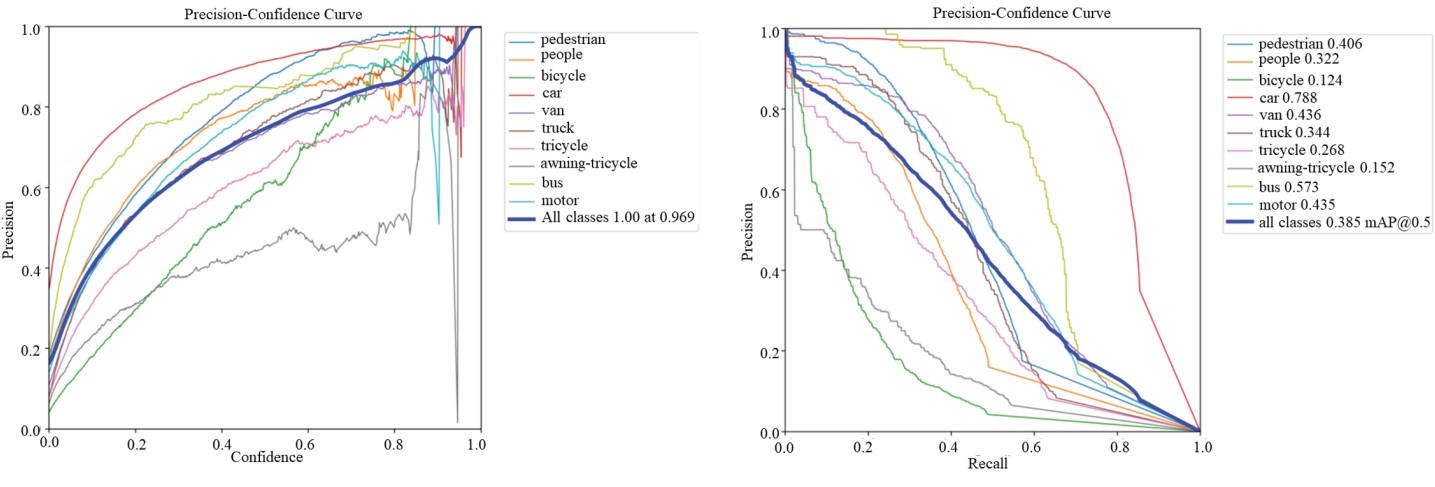

**Fig 4. Evalution factor of ours.**

shows that the improved model can detect more real targets while reducing missed detections.

**Comparison results with other versions of yolov8.** In order to verify the performance of the improved YOLOv8 model we proposed in different scenarios, we conducted comparative experiments on the VisDrone2019 validation set and test set. The experimental results are recorded in Tables 2 and 3, respectively, showing the performance of YOLOv8 models of different sizes and our proposed improved model in various indicators.

As can be seen from Table 2, the improved YOLOv8s outperforms other YOLOv8 models in most indicators. In terms of precision, the improved YOLOv8s reached 54.52%, higher than YOLOv8l's 54.22%; in terms of recall, the improved YOLOv8s reached 41.82%, slightly lower than YOLOv8l's 42.42%; in terms of F1-Score and mAP0.5, the improved YOLOv8s were 0.49 and 43.32% respectively, both of which were similar to YOLOv8l, but slightly lower in mAP0.5:0.9. In terms of detection time and model size, the improved YOLOv8s also showed better balance, with a detection time of 6.2ms and a model size of 21.6MB.

As can be seen from Table 3, the improved YOLOv8s also outperformed other YOLOv8 models on the test set. In terms of precision, the improved YOLOv8s reached 49.72%, higher than YOLOv8l's 48.12%; in terms of recall, the improved YOLOv8s reached 36.42%, lower than YOLOv8l's 37.2%; in terms of F1-Score and mAP0.5, the improved YOLOv8s were 0.44 and 35.22% respectively, slightly higher than YOLOv8l. In terms of mAP-0.5:0.9, the

**Table 2. Comparing results with other versions of yolov8 on validation.**

| Models | Recall (%) | Precision (%) | F1-Score | mAP0.5 (%) | mAP0.5:0.9 (%) | Detection Time (ms) | Model Size (MB) |
|---|---|---|---|---|---|---|---|
| YOLOv8n | 32.52 | 43.22 | 0.39 | 32.62 | 18.92 | 4.4 | 6.2 |
| YOLOv8s | 38.02 | 49.92 | 0.45 | 38.52 | 23.02 | 6.0 | 22.5 |
| YOLOv8m | 41.02 | 52.52 | 0.48 | 41.72 | 25.22 | 11.4 | 49.2 |
| YOLOv8l | 42.42 | 54.22 | **0.49** | 43.02 | **26.52** | 14.8 | 87.6 |
| PVswin-YOLOv8s [29] | 41.8 | 54.5 | 0.47 | 43.3 | - | 6.2 | 21.6 |
| Proposed | **41.82** | **54.52** | **0.49** | **43.32** | 26.42 | 6.2 | 21.6 |

**Table 3. Comparing results with other versions of yolov8 on testing.**

| Models | Recall (%) | Precision (%) | F1-Score | mAP0.5 (%) | mAP0.5:0.9 (%) | Detection Time (ms) | Model Size (MB) |
|---|---|---|---|---|---|---|---|
| YOLOv8n | 28.72 | 38.22 | 0.34 | 26.12 | 14.62 | 6.2 | 6.2 |
| YOLOv8s | 32.32 | 44.52 | 0.39 | 30.52 | 17.42 | 22.5 | 22.5 |
| YOLOv8m | 35.32 | 46.32 | 0.42 | 33.42 | 19.42 | 49.2 | 49.2 |
| YOLOv8l | 37.02 | 48.12 | 0.43 | 35.02 | **20.62** | 87.6 | 87.6 |
| Proposed | **36.42** | 49.72 | **0.44** | **35.22** | 20.42 | 21.6 | 21.6 |

improved YOLOv8s performed slightly lower. In terms of model size, the improved YOLOv8s maintained a relatively compact model structure with a size of 21.6MB.

The experimental results on the VisDrone2019 validation set and test set show that the improved YOLOv8 model we proposed performs well in most indicators, especially in key indicators such as precision and mAP0.5. At the same time, the improved YOLOv8s also shows a better balance in detection time and model size. Overall, our method has good efficiency and compactness while maintaining high detection performance, and is suitable for application in actual urban traffic monitoring systems.

**Comparison results with other versions of yolo.** To further verify the effectiveness of the improved YOLOv8 model we proposed, we conducted comparative experiments with other versions of the YOLO algorithm on the VisDrone2019 validation set (VisDrone2019-val). Table 4 shows the comparison results of different YOLO models, including the performance of YOLOv3 (tiny), YOLOv5, YOLOv6, YOLOv7, YOLOv8s, and improved YOLOv8s.

As can be seen from Table 4, the improved YOLOv8s outperforms other versions of the YOLO model in most indicators. In terms of precision, the improved YOLOv8s reached 54.52%, significantly higher than other models; in terms of recall, the improved YOLOv8s reached 41.82%, second only to YOLOv7's 41.12%. On mAP0.5 and mAP0.5:0.95, the improved YOLOv8s reached 43.32% and 26.42%, respectively, both of which are the best performances.

Although YOLOv7 performs best in detection time with 1.9ms, its model size is 72MB, which is much higher than the 21.6MB of the improved YOLOv8s. The improved YOLOv8s has a relatively small model size and a moderate detection time (6.2ms) while maintaining high detection performance, showing better balance.

By comparing the experimental results, it can be seen that the improved YOLOv8 model we proposed performs well in multiple key indicators, especially in important indicators such as precision and mAP0.5. Compared with other versions of the YOLO model, the improved YOLOv8s has better efficiency and compactness while maintaining high detection performance, and is suitable for application in actual urban traffic monitoring systems. This result

**Table 4. Comparison results with different versions of yolo.**

| Models | Recall (%) | Precision (%) | mAP0.5 (%) | mAP0.5:0.9 (%) | Detection Time (ms) | Model Size (MB) |
|---|---|---|---|---|---|---|
| YOLOv3 (tiny) | 24.12 | 37.22 | 23.22 | 12.92 | 2.7 | 24 |
| YOLOv5 | 31.92 | 42.32 | 31.52 | 18.02 | 3.8 | **5.3** |
| YOLOv6 | 29.42 | 39.82 | 29.12 | 17.02 | 3.3 | 8.7 |
| YOLOv7 | 41.12 | 50.22 | 37.92 | 19.92 | **1.9** | 72 |
| YOLOv8s | 38.02 | 49.92 | 38.52 | 23.02 | 6.0 | 22.5 |
| Ours | **41.82** | **54.52** | **43.32** | **26.42** | 6.2 | 21.6 |

proves the effectiveness of our improvement strategy and provides a valuable reference for future object detection research.

**Comparison results with other classical methods.** To fully verify the effectiveness of the improved YOLOv8 model we proposed, we conducted comparative experiments with some classic object detection models. Table 5 shows the performance of these models in mAP0.5 and mAP0.5:0.95 on the VisDrone2019 validation set.

As can be seen from Table 5, the improved YOLOv8s outperforms other classic models in both mAP0.5 and mAP0.5:0.95. In terms of mAP0.5, the improved YOLOv8s reached 43.32%, significantly higher than Cascade R-CNN's 39.12% and Faster R-CNN's 37.2%. In terms of mAP0.5:0.95, the improved YOLOv8s also performed well, reaching 26.42%, significantly better than Cascade R-CNN's 24.32% and Faster R-CNN's 21.92%.

The experimental results of the comparison with the classic model show that the improved YOLOv8 model we proposed performs well in target detection, especially in key indicators such as mAP0.5 and mAP0.5:0.95. This result not only proves the effectiveness of our improvement strategy, but also shows that the improved YOLOv8s has higher detection accuracy and robustness in practical applications. Our model can provide more accurate and reliable detection results in complex scenes, providing strong technical support for urban traffic monitoring systems.

**Day vs. Night.** In addition, we also compared the detection results during the day and at night, as shown in Fig 5. It can be seen that at night, due to the influence of external factors such as light, the detection effect of the model has declined to a certain extent. In the daytime, due to sufficient light, the detection performance also improves.

## 4.4 Ablation study

For the effectiveness of our proposed improved YOLOv8 model, we conduct ablation experiments and compared them on the VisDrone2019 validation set (VisDrone2019-val). Table 6

**Table 5. Comparison with some classical models.**

| Models | mAP0.5 (%) | mAP0.5:0.95 (%) |
|---|---|---|
| Cascade RCNN | 37.22 | 21.92 |
| Faster R-CNN | 39.12 | 24.32 |
| RetinaNet | 19.12 | 10.62 |
| CenterNet [30] | 33.72 | 18.82 |
| Ours | **43.32** | **26.42** |

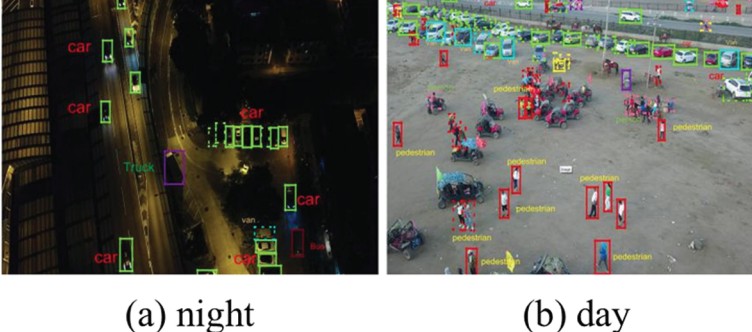

**Fig 5. Comparison of test results during daytime and nighttime.**

**Table 6. Ablation Study results, where ↑ shows an increment in results.**

| Models | mAP0.5 (%) | mAP0.5:0.95 (%) |
|---|---|---|
| Baseline YOLOv8s | 38.52 | 23.02 |
| YOLOv8s + MFF | 40.62 (↑ 2.12%) | 24.72 (↑ 1.72%) |
| YOLOv8s + MFF+INMS | 42.22 (↑ 1.32%) | 25.62 (↑0.92%) |
| Ours | **43.32** (↑ 1.12%) | **26.42** (↑ 0.82%) |

shows the impact of different module combinations on model performance, where our method is compared with modules of different combinations.

As can be seen from Table 6, the model performance has been significantly improved after adding different modules. The mAP0.5 of the baseline YOLOv8s is 38.5%, and the mAP0.5:0.95 is 23.02%. After adding module one, the mAP0.5 of the improved YOLOv8s is increased to 40.62%, and the mAP0.5:0.95 is increased to 24.72%, an increase of 2.12% and 1.72%, respectively. After adding modules one and two, the mAP0.5 of the improved YOLOv8s is further increased to 42.22%, and the mAP0.5:0.95 is increased to 25.62%, an increase of 1.32% and 0.92%, respectively.

Finally, the complete improved YOLOv8s model achieved 43.32% on mAP0.5 and 26.42% on mAP0.5:0.95, an increase of 1.12% and 0.82%, respectively. These results show that by introducing modules one and two, our method has significantly improved the performance of object detection.

The ablation experiment results show that the performance of our proposed improved YOLOv8 model has been significantly improved after adding different modules. Our method performs well in key indicators such as mAP0.5 and mAP0.5:0.95, significantly outperforming the baseline YOLOv8s model. This result proves the effectiveness of our proposed module design and provides a valuable reference for future object detection research. Our improved YOLOv8s model can provide more accurate and reliable detection results in complex scenes, providing strong technical support for urban traffic monitoring systems.

## 4.5 Visualization results

To deeply analyze the performance of our method in object detection, we generated a confusion matrix and compared it with the original YOLOv8 model. Fig 6 shows the confusion matrices of the two models. By confusion matrices, we can get a detailed understanding of the prediction effect and false detection of each category.

1. Pedestrian: In the confusion matrix of YOLOv8s, the correct detection rate of pedestrians is 33%, while the correct detection rate of the improved YOLOv8s is increased to 39%.
   In YOLOv8s, the proportion of pedestrians being misdetected as background is 64%, while in the improved model, this proportion is reduced to 58%. This shows that the improved model performs better in reducing false detection of pedestrians.
2. Car: The correct detection rate of YOLOv8s on the vehicle category is 73%, while the correct detection rate of the improved YOLOv8s is increased to 75%. The improved model also performs well in vehicle detection.
   In the case of misdetection as background, the false detection rate of YOLOv8s is 25%, while the improved model is reduced to 22%.

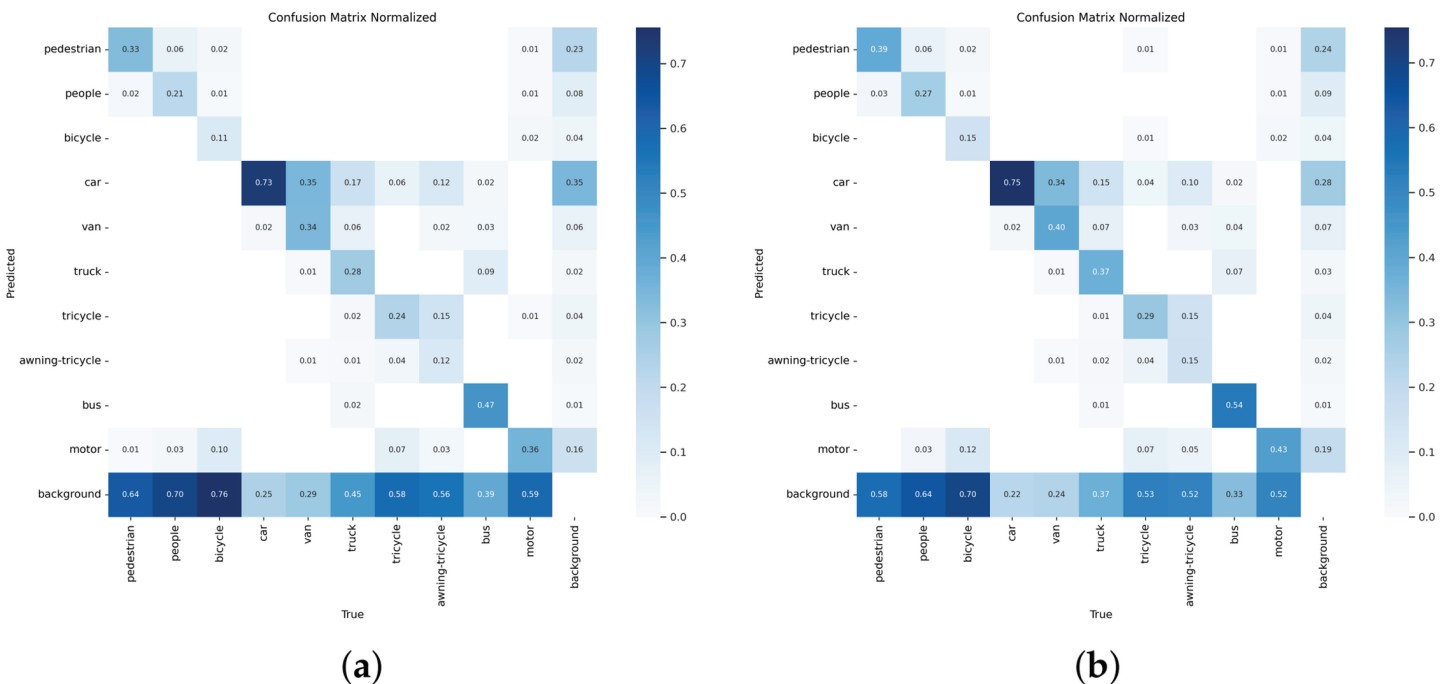

**Fig 6. (a) YOLOv8s confusion matrix (b) Our confusion matrix.**

3. Bicycle: The correct detection rate of YOLOv8s on the bicycle category is 11%, while the correct detection rate of the improved YOLOv8s is increased to 15%. The improved model has significantly improved its performance in bicycle detection.
   When the false detection is background, the false detection rate of YOLOv8s is 76%, while the improved model reduces it to 70%.

4. Motorcycle (Motor): The correct detection rate of YOLOv8s in the motorcycle category is 36%, while the correct detection rate of the improved YOLOv8s is increased to 43%. The improved model also performs well in motorcycle detection.
   When the false detection is background, the false detection rate of YOLOv8s is 39%, while the improved model reduces it to 33%.

5. Bus: The correct detection rate of YOLOv8s in the bus category is 47%, while the correct detection rate of the improved YOLOv8s is increased to 54%. The improved model performs well in bus detection.
   When the false detection is background, the false detection rate of YOLOv8s is 2%, while the improved model reduces it to 1%.

By comparing the confusion matrix, the improved YOLOv8 model has shown higher accuracy and lower false detection rate in the detection of each category. This shows that the target detection performance of the improved model in complex scenes has been significantly improved, and it can more accurately identify and classify various targets and reduce false detection and missed detection.

The comparison results of the confusion matrix further verify the effectiveness of the improved YOLOv8 model we proposed. In the main target categories such as pedestrians, vehicles, bicycles, motorcycles and buses, the improved model has shown higher detection accuracy and lower false detection rate. This result proves the superior performance of the

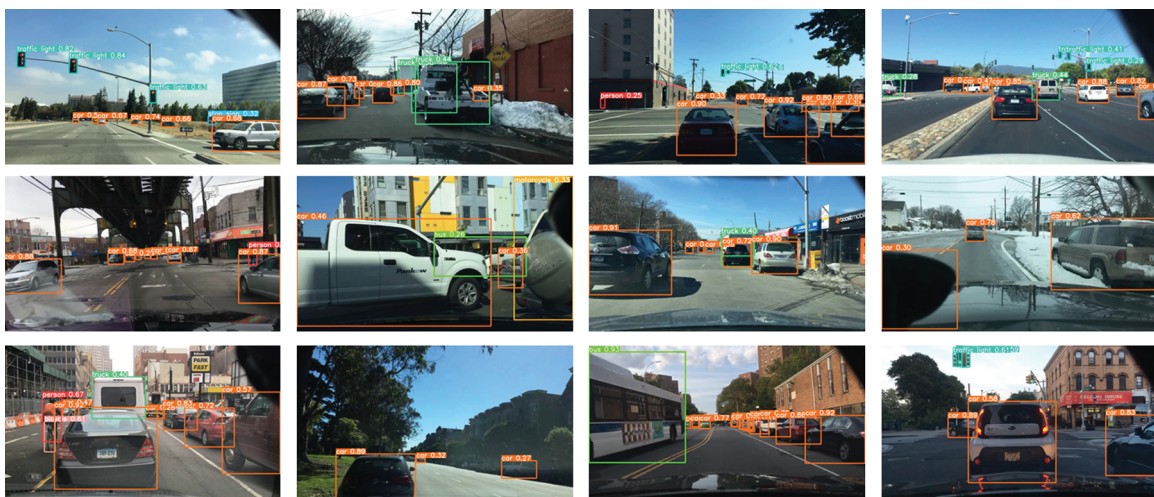

**Fig 7. Visualization results of our approach.**

improved YOLOv8 model in practical applications and provides a more reliable and accurate solution for urban traffic monitoring systems.

To verify the effectiveness of the improved YOLOv8 model we proposed in practical applications, we conducted a variety of detection tasks and visualized the results. By observing the detection results of urban traffic scenes, pedestrian-dense areas and highway scenes, our method performs very well in various scenes.

In urban traffic scenes, our method can accurately identify and annotate traffic lights and vehicles, demonstrating superior detection performance in complex traffic environments, The results are shown in Fig 7. In the crowded urban street scene, our method successfully detects multiple pedestrians and accurately labels each pedestrian, demonstrating efficient detection capabilities in crowded environments. In the multi-target detection task in urban traffic, our method can simultaneously identify and label multiple categories of targets, such as vehicles, pedestrians, and bicycles, demonstrating its outstanding performance in multi-target detection tasks. In the highway scene, our method accurately identifies and labels trucks and vehicles, demonstrating its potential for application in highway environments.

Overall, the visualization results of these detection tasks fully demonstrate the effectiveness and superiority of our proposed improved YOLOv8 model. Whether in complex traffic environments or on streets with dense pedestrians, our method can provide high-precision and high-reliability target detection, providing strong technical support for urban traffic monitoring systems. These results show that our model not only performs well in experiments but also has wide applicability and significant performance improvements in practical applications.

## 5 Conclusion

In this study, we proposed an object detection model based on improved YOLOv8. By introducing a MFF module and an INMS algorithm, the detection performance was significantly improved. The MFF module enables the model to better detect objects of different sizes by combining feature maps of different scales; the INMS algorithm effectively reduces repeated detection and missed detection by optimizing the screening process of candidate boxes. Through experimental verification on the VisDrone2019 dataset, our improved model outperforms other classic models and the baseline YOLOv8 model in all indicators, especially

in accuracy and mean average precision (mAP). The visualization effect diagram further proves the superiority of our method in practical applications. In various complex urban traffic scenes, our improved model can accurately identify and annotate multiple categories of objects, including traffic lights, vehicles, and pedestrians, showing excellent multi-target detection capabilities. In short, the improved YOLOv8 model proposed in this study not only performs well in experiments, but also has significant performance improvements in practical applications. Our model can provide high-precision and high-reliability target detection for urban traffic monitoring systems, which has important practical application significance and broad development prospects. This research result provides strong support for future research on target detection technology and the application of urban traffic monitoring systems.

Although our method has achieved good results, we believe that it is still necessary to discuss its limitations. We have not conducted corresponding experimental tests for real-world application scenarios, which is our shortcoming. In future work, we believe that it is worthwhile to increase the application exploration of real-world scenarios, which can also promote further optimization of our model.

## Author contributions

**Conceptualization:** Huili Dou.

**Data curation:** Huili Dou, Sirui Chen.

**Resources:** Yuanyuan Liu.

**Software:** Fangyuan Xu.

**Supervision:** Huili Dou, Fangyuan Xu.

**Validation:** Yuanyuan Liu.

**Visualization:** Hongyang Zhao.

**Writing – review & editing:** Hongyang Zhao.

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
