## [Decision Letter · Decision Letter 0]

7 Aug 2024

PONE-D-24-29268Analysis of the effect of using the improved YOLOv8 model for vehicle and pedestrian detection in urban traffic monitoring systemPLOS ONE

Dear Dr. Dou,

Thank you for submitting your manuscript to PLOS ONE. After careful consideration, we feel that it has merit but does not fully meet PLOS ONE’s publication criteria as it currently stands. Therefore, we invite you to submit a revised version of the manuscript that addresses the points raised during the review process.

We look forward to receiving your revised manuscript.

Kind regards,

Khalil Abdelrazek Khalil, Ph.D.

Academic Editor

PLOS ONE

Journal Requirements:

 "a.This work was sponsored in part by Zhejiang Provincial Natural Science Foundation of China under Grant No. LTGG24E080005: Research on Data-Driven Service Performance Assessment and Maintenance Decision-Making of Rural Highway Bridge Groups."

4. In this instance it seems there may be acceptable restrictions in place that prevent the public sharing of your minimal data. However, in line with our goal of ensuring long-term data availability to all interested researchers, PLOS’ Data Policy states that authors cannot be the sole named individuals responsible for ensuring data access (http://journals.plos.org/plosone/s/data-availability#loc-acceptable-data-sharing-methods).

Reviewers' comments:

Reviewer's Responses to Questions

**Comments to the Author**

1. Is the manuscript technically sound, and do the data support the conclusions?

Reviewer #1: Yes

Reviewer #2: Yes

2. Has the statistical analysis been performed appropriately and rigorously? 

Reviewer #1: Yes

Reviewer #2: Yes

3. Have the authors made all data underlying the findings in their manuscript fully available?

Reviewer #1: Yes

Reviewer #2: Yes

4. Is the manuscript presented in an intelligible fashion and written in standard English?

Reviewer #1: Yes

Reviewer #2: Yes

5. Review Comments to the Author

Reviewer #1: 1. The methodology section provides detailed descriptions of the improvements made to the YOLOv8 model, including multi-scale feature fusion and the improved non-maximum suppression (NMS) algorithm. However, it would be beneficial to include a flowchart or diagram summarizing the steps of the proposed method to enhance clarity and ensure consistency in understanding the proposed improvements.

2. The manuscript includes several evaluation metrics such as precision, recall, F1-score, mAP0.5, and mAP0.5:0.95. While these metrics are comprehensive, the discussion could benefit from a deeper analysis of the trade-offs between these metrics, particularly in scenarios with dense traffic and occlusions.

3. The manuscript compares the improved YOLOv8 model with several state-of-the-art models. However, the comparison could be more robust by including additional recent models that have shown significant performance in similar tasks.

4. The ablation study effectively demonstrates the contributions of individual components of the improved YOLOv8 model. However, the manuscript would benefit from a more detailed discussion on the impact of each component on the overall model performance.

5. The manuscript claims that the improved YOLOv8 model provides reliable technical support for urban traffic monitoring systems. While the experimental results are promising, the discussion on real-world applicability and deployment challenges is limited.

6. he manuscript provides quantitative results but lacks sufficient visual examples or case studies that illustrate the improved detection capabilities of the YOLOv8 model.

Reviewer #2: This paper presents an improved YOLOv8 model for vehicle and pedestrian detection in urban traffic monitoring systems. In order to improve the detection performance of the model, introduced a multi-scale feature fusion module and an improved non-maximum suppression (NMS) algorithm based on the YOLOv8 model. The multi-scale feature fusion module enhances the model’s detection ability for targets of different sizes by combining feature maps of different scales; the improved non-maximum suppression algorithm effectively reduces repeated detection and missed detection by optimizing the screening process of candidate boxes. Experimental results show that the improved YOLOv8 model exhibits excellent detection performance on the VisDrone2019 dataset, and outperforms other classic target detection models and the baseline YOLOv8 model in key indicators such as precision, recall, F1 score, and mean average precision (mAP).

The manuscript needs improvement; therefore, I recommend minor revision. The detailed comments are given below. 

1)The title of the paper is not impressive; it should be renamed. Additionally, for the experimental work, VisDrone 2019 dataset was used which means this paper is based on UAV technology. Therefore, authors should link this technology in the title for the reader's clear understanding.

2)There is no numbering with each heading and the sub-sections of each heading are not easily understandable. For example, at the end of the related work, there are other methods and methods sessions that do not give proper understanding to the new readers. Furthermore, the initial letters of each word in the sub-session in related work start from small letters, while in the other headings, the first letter of each word in the sub-session starts from a capital letter. It is necessary to set the whole paper format.

3)How the proposed algorithms play an important role in other issues such as occlusion, low-quality images, multispectral images, and joint issues for target detection in traffic congestion using UAVs and AVs as mentioned in the following papers:

Target Detection and Recognition for Traffic Congestion in Smart Cities Using Deep Learning-Enabled UAVs: A Review and Analysis

Deep Learning-Based Pedestrian Detection in Autonomous Vehicles: Substantial Issues and Challenges

Advance generalization technique through 3D CNN to overcome the false positives pedestrian in autonomous vehicles

Please justify the positive aspects of the proposed algorithm on other issues for target detection in traffic congestion and also include the mentioned references in it.

4)The author considers the VisDrone2019 dataset rather than the other datasets such as UAVDT, VisioDECT, and Synthetic Drone datasets. Why are the other datasets not considerable for this work? Please elaborate on it.

5)In the results and analysis section, the authors compare the results of the proposed model with other versions but the heading of comparison doesn’t make sense such as it is “Comparison results with other versions of yolov8”, it should be renamed such as “Comparison of the results with other versions of yolov8’ or “Comparing results with other versions of yolov8”.

6)In the introduction and experiment section, pages number 74, 390, 423, 425, and 428; the authors mentioned that we used two public datasets in our experimental work but this paper discusses only VisDrone dataset. What are the brief introduction and experimental results of the second public dataset? If this paper uses only one dataset on the entire paper, then it should be corrected on page number 74, 390, 423, 425, and 428.

7)The VisDrone dataset covers different weather conditions (such as sunny, cloudy, rainy) and time periods (such as day, dusk, and night). The authors need to justify the model results on the weather conditions and time periods. Is the model performing best in the sunny, cloudy, rainy, and daytime or either at nighttime? In which condition is the result very effective?

6. PLOS authors have the option to publish the peer review history of their article (what does this mean?). If published, this will include your full peer review and any attached files.

Reviewer #1: No

Reviewer #2: **Yes: **Muhammad Asim

---

## [Author Response · Author response to Decision Letter 1]

26 Aug 2024

Reviewer #1:

1. The methodology section provides detailed descriptions of the improvements made to the YOLOv8 model, including multi-scale feature fusion and the improved non-maximum suppression (NMS) algorithm. However, it would be beneficial to include a flowchart or diagram summarizing the steps of the proposed method to enhance clarity and ensure consistency in understanding the proposed improvements.

Response: Thank you for your comments, which made us realize the shortcomings of our work. We have described your comments in the article. For example, the network structure shown in Figure 2 is the detailed process of the module, which we hope can improve the understanding of the article.

2. The manuscript includes several evaluation metrics such as precision, recall, F1-score, mAP0.5, and mAP0.5:0.95. While these metrics are comprehensive, the discussion could benefit from a deeper analysis of the trade-offs between these metrics, particularly in scenarios with dense traffic and occlusions.

Response: Thank you for your comments, which made us realize the shortcomings of our work. We have described your comments in the article. In the results and analysis of the experiment section, we described in detail the results of the method on various indicators and the relationship between them. For details, please see the red marked content in the results and analysis.

3. The manuscript compares the improved YOLOv8 model with several state-of-the-art models. However, the comparison could be more robust by including additional recent models that have shown significant performance in similar tasks.

Response: Thank you for your comments, which made us aware of the shortcomings of our work. We have added some experimental results of recent articles for comparison in the experimental section, see Table 2 for details.

4. The ablation study effectively demonstrates the contributions of individual components of the improved YOLOv8 model. However, the manuscript would benefit from a more detailed discussion on the impact of each component on the overall model performance.

Response: Thank you for your comments, which made us aware of the shortcomings of our work. We have conducted relevant experiments in ablation experiments to investigate the impact of model components on model performance. For details, please refer to the ablation experiment section.

5. The manuscript claims that the improved YOLOv8 model provides reliable technical support for urban traffic monitoring systems. While the experimental results are promising, the discussion on real-world applicability and deployment challenges is limited.

Response: Thank you for your comments, which made us realize the shortcomings of our work. Your comments are the limitations of our model, which we have described in the conclusion section and the improvement plan for future work. Please see the red marked content in the conclusion section for details.

6. The manuscript provides quantitative results but lacks sufficient visual examples or case studies that illustrate the improved detection capabilities of the YOLOv8 model.

Response: Thank you for your comments, which have helped us to further improve our work. In response to your comments, we have shown the detection results of our method in the visualization results of the experimental part, hoping to better reflect the effectiveness and practicality of our method.

Reviewer #2: This paper presents an improved YOLOv8 model for vehicle and pedestrian detection in urban traffic monitoring systems. In order to improve the detection performance of the model, introduced a multi-scale feature fusion module and an improved non-maximum suppression (NMS) algorithm based on the YOLOv8 model. The multi-scale feature fusion module enhances the model’s detection ability for targets of different sizes by combining feature maps of different scales; the improved non-maximum suppression algorithm effectively reduces repeated detection and missed detection by optimizing the screening process of candidate boxes. Experimental results show that the improved YOLOv8 model exhibits excellent detection performance on the VisDrone2019 dataset, and outperforms other classic target detection models and the baseline YOLOv8 model in key indicators such as precision, recall, F1 score, and mean average precision (mAP).

The manuscript needs improvement; therefore, I recommend minor revision. The detailed comments are given below.

1)The title of the paper is not impressive; it should be renamed. Additionally, for the experimental work, VisDrone 2019 dataset was used which means this paper is based on UAV technology. Therefore, authors should link this technology in the title for the reader's clear understanding.

Response: Thank you for your comments, which have helped us further improve our work. We have made serious revisions based on your comments, modified the title of the article, and added drone content to make it more in line with the content of the article.

2)There is no numbering with each heading and the sub-sections of each heading are not easily understandable. For example, at the end of the related work, there are other methods and methods sessions that do not give proper understanding to the new readers. Furthermore, the initial letters of each word in the sub-session in related work start from small letters, while in the other headings, the first letter of each word in the sub-session starts from a capital letter. It is necessary to set the whole paper format.

Response: Thank you for your comments, which have helped us to further improve our work. According to your comments, we have revised the title of the article as a whole to make it easier to understand.

3)How the proposed algorithms play an important role in other issues such as occlusion, low-quality images, multispectral images, and joint issues for target detection in traffic congestion using UAVs and AVs as mentioned in the following papers:

•Target Detection and Recognition for Traffic Congestion in Smart Cities Using Deep Learning-Enabled UAVs: A Review and Analysis

•Deep Learning-Based Pedestrian Detection in Autonomous Vehicles: Substantial Issues and Challenges

• Advance generalization technique through 3D CNN to overcome the false positives pedestrian in autonomous vehicles

Please justify the positive aspects of the proposed algorithm on other issues for target detection in traffic congestion and also include the mentioned references in it.

Response: Thank you for your comments, which have helped us further improve our work. By comparing with relevant literature, we further elaborate on the active handling of occluded and masked objects by our model. For details, please see the red marked part in Section 2.3.

4)The author considers the VisDrone2019 dataset rather than the other datasets such as UAVDT, VisioDECT, and Synthetic Drone datasets. Why are the other datasets not considerable for this work? Please elaborate on it.

Response: We understand your confusion about this issue. Compared with datasets such as UAVDT, VisioDECT, and Synthetic Drone, VisDrone2019 is a more challenging dataset. Its complex surrounding environment and the occlusion and cover between objects virtually increase the difficulty of object detection and also put higher requirements on the performance of the model. Therefore, we choose VisDrone2019 dataset as our first choice.

5)In the results and analysis section, the authors compare the results of the proposed model with other versions but the heading of comparison doesn’t make sense such as it is “Comparison results with other versions of yolov8”, it should be renamed such as “Comparison of the results with other versions of yolov8’ or “Comparing results with other versions of yolov8”.

Response: Thank you for your comments, which have helped us improve our work. We have modified the title of the experimental results based on your comments.

6)In the introduction and experiment section, pages number 74, 390, 423, 425, and 428; the authors mentioned that we used two public datasets in our experimental work but this paper discusses only VisDrone dataset. What are the brief introduction and experimental results of the second public dataset? If this paper uses only one dataset on the entire paper, then it should be corrected on page number 74, 390, 423, 425, and 428.

Response: Thank you for your comments, which have helped us improve our work. In this work, we only used one dataset to validate the model, and we have corrected the errors in the paper.

7)The VisDrone dataset covers different weather conditions (such as sunny, cloudy, rainy) and time periods (such as day, dusk, and night). The authors need to justify the model results on the weather conditions and time periods. Is the model performing best in the sunny, cloudy, rainy, and daytime or either at nighttime? In which condition is the result very effective?

Response: Thank you for your comments, which have helped us improve our work. We have also conducted an additional experimental comparison between daytime and nighttime. The results show that the detection effect at night is somewhat reduced compared to that during the daytime due to external factors such as lighting.

---

## [Decision Letter · Decision Letter 1]

18 Nov 2024

Analysis of vehicle and pedestrian detection effects of improved YOLOv8 model in drone-assisted urban traffic monitoring system

PONE-D-24-29268R1

Dear Dr. Dou,

We’re pleased to inform you that your manuscript has been judged scientifically suitable for publication and will be formally accepted for publication once it meets all outstanding technical requirements.

Kind regards,

Khalil Abdelrazek Khalil, Ph.D.

Academic Editor

PLOS ONE

Additional Editor Comments (optional):

Reviewers' comments:

Reviewer's Responses to Questions

**Comments to the Author**

1. If the authors have adequately addressed your comments raised in a previous round of review and you feel that this manuscript is now acceptable for publication, you may indicate that here to bypass the “Comments to the Author” section, enter your conflict of interest statement in the “Confidential to Editor” section, and submit your "Accept" recommendation.

Reviewer #1: All comments have been addressed

Reviewer #2: All comments have been addressed

2. Is the manuscript technically sound, and do the data support the conclusions?

Reviewer #1: Yes

Reviewer #2: Yes

3. Has the statistical analysis been performed appropriately and rigorously? 

Reviewer #1: Yes

Reviewer #2: Yes

4. Have the authors made all data underlying the findings in their manuscript fully available?

Reviewer #1: Yes

Reviewer #2: Yes

5. Is the manuscript presented in an intelligible fashion and written in standard English?

Reviewer #1: Yes

Reviewer #2: Yes

6. Review Comments to the Author

Reviewer #1: Authors revised the manuscript based on my comments and I am satisfied with the manuscript and its in publishable form

Reviewer #2: Authors have addressed all my concerns in the revised version of the manuscript. It can be accepted for possible publication.

7. PLOS authors have the option to publish the peer review history of their article (what does this mean?). If published, this will include your full peer review and any attached files.

Reviewer #1: No

Reviewer #2: **Yes: **Muhammad Asim

---

## [Editor Report · Acceptance letter]

PONE-D-24-29268R1

PLOS ONE

Dear Dr. Dou,

I'm pleased to inform you that your manuscript has been deemed suitable for publication in PLOS ONE. Congratulations! Your manuscript is now being handed over to our production team.

Kind regards,

on behalf of

Dr. Khalil Abdelrazek Khalil

Academic Editor

PLOS ONE